# Down-Regulation of CYP3A4 by the K_Ca_1.1 Inhibition Is Responsible for Overcoming Resistance to Doxorubicin in Cancer Spheroid Models

**DOI:** 10.3390/ijms242115672

**Published:** 2023-10-27

**Authors:** Susumu Ohya, Junko Kajikuri, Hiroaki Kito, Miki Matsui

**Affiliations:** Department of Pharmacology, Graduate School of Medical Sciences, Nagoya City University, Nagoya 467-8601, Japan; kajikuri@med.nagoya-cu.ac.jp (J.K.); kito@med.nagoya-cu.ac.jp (H.K.); miki.matsui.4238@gmail.com (M.M.)

**Keywords:** cancer spheroid, doxorubicin resistance, Ca^2+^-activated K^+^ channel, K_Ca_1.1, Akt-Nrf2, CYP3A4

## Abstract

The large-conductance Ca^2+^-activated K^+^ channel, K_Ca_1.1, plays a pivotal role in cancer progression, metastasis, and the acquisition of chemoresistance. Previous studies indicated that the pharmacological inhibition of K_Ca_1.1 overcame resistance to doxorubicin (DOX) by down-regulating multidrug resistance-associated proteins in the three-dimensional spheroid models of human prostate cancer LNCaP, osteosarcoma MG-63, and chondrosarcoma SW-1353 cells. Investigations have recently focused on the critical roles of intratumoral, drug-metabolizing cytochrome P450 enzymes (CYPs) in chemoresistance. In the present study, we examined the involvement of CYPs in the acquisition of DOX resistance and its overcoming by inhibiting K_Ca_1.1 in cancer spheroid models. Among the CYP isoforms involved in DOX metabolism, CYP3A4 was up-regulated by spheroid formation and significantly suppressed by the inhibition of K_Ca_1.1 through the transcriptional repression of CCAAT/enhancer-binding protein, CEBPB, which is a downstream transcription factor of the Nrf2 signaling pathway. DOX resistance was overcome by the siRNA-mediated inhibition of CYP3A4 and treatment with the potent CYP3A4 inhibitor, ketoconazole, in cancer spheroid models. The phosphorylation levels of Akt were significantly reduced by inhibiting K_Ca_1.1 in cancer spheroid models, and K_Ca_1.1-induced down-regulation of CYP3A4 was reversed by the treatment with Akt and Nrf2 activators. Collectively, the present results indicate that the up-regulation of CYP3A4 is responsible for the acquisition of DOX resistance in cancer spheroid models, and the inhibition of K_Ca_1.1 overcame DOX resistance by repressing CYP3A4 transcription mainly through the Akt-Nrf2-CEBPB axis.

## 1. Introduction

Three-dimensional (3D) cancer spheroid models using ultra-low attachment plates and dishes form spherical self-assembled aggregates of cancer cells and are a valuable tool for examining the tumor microenvironment (TME) in solid cancers with the characteristics of cancer stem cells (CSCs), such as a high metastatic capacity and chemoresistance [1]. Recent studies indicated the essential role of K^+^ channels in cancer cell proliferation, invasion, migration, and metastasis [2,3]. Among Ca^2+^-activated K^+^ channel superfamily members (large-conductance K_Ca_1.1, small-conductance K_Ca_2.1–2.3, and intermediate-conductance K_Ca_3.1), K_Ca_1.1, encoded by KCNMA1, contributes to cancer development, and its amplification positively correlated with high cancer stage, a high histological grade, and poor prognosis [2,3]. On the other hand, K_Ca_1.1 protein expression and channel activity were increased by 3D spheroid formation in osteosarcoma MG-63, chondrosarcoma SW-1353, and prostate cancer LNCaP cells, mediating the down-regulation of the E3 ubiquitin ligase, FBXW7 [4,5,6]. Resistance to chemotherapies, including doxorubicin (DOX), acquired by spheroid formation was significantly overcome by the inhibition of K_Ca_1.1 in their spheroid models [4,5]. In studies that focused on multidrug resistance (MDR/ABCB) and multidrug resistance-associated proteins (MRP/ABCC) belonging to subfamilies B and C in the ABC transporter superfamily, respectively, an increase in particular MRP proteins (MRP1 and MRP5) contributed to chemoresistance by spheroid formation, and their expression was down-regulated by the inhibition of K_Ca_1.1 [4,5].

Drug-metabolizing cytochrome P450 enzymes (CYPs) are responsible for the majority of phase I drug metabolism reactions and are present at high levels in the liver and intestines [7]. Many drugs and substances induce or down-regulate the expression of CYP isoforms [7]. Recent studies reported that some CYP isoforms are overexpressed in cancer cells, and their expression and activity levels are considered for the effective use of chemotherapeutics [8,9,10]. The expression levels of CYP isoforms correlate with differential therapeutic responses in cancer treatments. Hepatocellular carcinoma cell lines with higher levels of CYP3A4 exhibited an increase in DOX resistance, while those with lower levels of CYP3A4 showed a decrease in DOX resistance [11]. In TME, an increase in CYP3A4 metabolic activity plays an important role in the acquisition of chemoresistance by cancer cells [12]. These findings suggest the therapeutic potential of drugs that inhibit or down-regulate CYP3A4 to overcome DOX resistance and reduce DOX toxicity when used in combination.

DOX is one of the most widely used drugs in chemotherapy for the treatment of different types of cancers [13] and is a major substrate of CYP3A4-mediated oxidation [14]. The five main DOX metabolites in vivo are known: DOXol, DOX-semiquinone, DOX-hydroxyaglycone, DOX deoxyaglycone, and DOXol aglycone. Other CYP isoforms, such as CYP2B6 and CYP1B1, also contribute to DOX metabolism to a minor extent. The antimicrobial, ketoconazole, is a well-known potent CYP3A4 inhibitor [15]. The activation of the mitogen-activated protein kinase (MAPK)/extracellular signal-regulated kinase (ERK) and phosphatidyl inositol 3-kinase (PI3K)/Akt signaling pathways contributes to DOX resistance [16]. The activation of nuclear factor erythroid 2-related factor 2 (Nrf2) is also associated with the acquisition of DOX resistance through PI3K/Akt/Nrf2 and ERK/Nrf2 axes [17]. Furthermore, the PI3K/Akt signaling pathway is activated in CSCs, including sarcomas [18,19].

As described above, we previously reported the contribution of MRPs to the overcoming of DOX resistance through the inhibition of K_Ca_1.1 in K_Ca_1.1-expressing cancer spheroid models; however, the role of CYPs remains unclear. Therefore, we herein examined (1) the identification of CYP isoform(s) contributing to DOX resistance in MG-63, SW-1353, and LNCaP spheroid models, (2) the K_Ca_1.1 inhibition-induced down-regulation of their expression, and (3) the signaling mechanisms underlying the CYP-mediated overcome of DOX resistance through the inhibition of K_Ca_1.1.

## 2. Results

### 2.1. Overexpression of CYP2B6 and CYP3A4 Transcripts in LNCaP, MG-63, and SW-1353 Spheroid Models

In our previous studies, we showed functional expression of K_Ca_1.1 in isolated cells from 3D spheroids of LNCaP, MG-63, and SW-1353 using voltage-dye imaging and whole-cell patch clamp recordings [4,5]. In addition, intracellular Ca^2+^ rise was observed together with K_Ca_1.1 activation-induced hyperpolarization in them [5]. Similar to the previous reports [4,5], we initially confirmed the higher expression levels of Nanog and KLF4 in 3D spheroid models of K_Ca_1.1-expressing LNCaP, MG-63, and SW-1353 than those in two-dimensional (2D) monolayers by a real-time PCR assay.

To identify the CYP isoforms involved in DOX metabolism in 3D spheroid models, we examined the expression levels of CYP3A4, CYP1B1, CYP2A6, CYP2A7, CYP3A5, CYP2B6, CYP2C8, CYP2D6, and CYP4Z1 transcripts using real-time PCR. The expression levels of CYP3A4 and CYP2B6 were significantly increased in the 3D spheroid models of LNCaP (Figure 1A,B), MG-63 (Figure 1C,D), and SW-1353 (Figure 1E,F) (*n* = 4 for each, *p* < 0.01) (Figure 1). In contrast, the expression of the other CYP isoforms was very low, with expression levels < 0.005 in arbitrary units (a.u.) normalized to ACTB levels.

### 2.2. Overcoming DOX Resistance by Both the siRNA-Mediated and Pharmacological Inhibition of CYP3A4 in Cancer Spheroid Models

To elucidate the contribution of CYP3A4 and CYP2B6 to DOX resistance in the three cancer spheroid models [4,5], we examined the effects of their siRNA-mediated inhibition on DOX resistance. As shown in Figure 2, the siRNA-mediated inhibition of CYP3A4 reversed DOX resistance in LNCaP (Figure 2A), MG-63 (Figure 2B), and SW-1353 (Figure 2C) spheroid models (*n* = 5, *p* < 0.01), whereas no significant changes in DOX sensitivity were observed following the inhibition of CYP2B6 (*n* = 5, *p* > 0.05) (Figure 2A–C). No significant changes in cell viability were noted following transfection of siRNAs alone (Appendix A) (*n* = 5, *p* > 0.05) and treatment with DOX without siRNAs (Appendix A) (*n* = 5, *p* > 0.05). The inhibition efficacy of siRNA-mediated target genes was approximately 50% for each (Appendix A). Consistent with the results obtained on the siRNA-mediated inhibition of CYP3A4, its pharmacological inhibition with the potent CYP3A4 inhibitor, ketoconazole (KCZ, 1 μM), significantly reversed DOX resistance in cancer spheroid models (*n* = 5, *p* < 0.01) (Figure 2D–F). No significant changes in cell viability were noted following a single treatment with 1 μM KCZ for 48 h (Appendix A). The concentration–response relationships of DOX and KCZ of 2D- and 3D-cultured cancer spheroid models were shown in Appendix A. These results strongly suggest that CYP3A4 is the main CYP isoform contributing to DOX resistance acquired by cancer spheroid formation.

### 2.3. Transcriptional Repression of CYP3A4 by the Inhibition of K_Ca_1.1 in Cancer Spheroid Models

Previous studies showed that the inhibition of K_Ca_1.1 with the potent and selective K_Ca_1.1 inhibitor, paxilline (PAX), overcame DOX resistance in MG-63, SW-1353, and LNCaP spheroid models [4,5]. To elucidate the contribution of CYP3A4 to the K_Ca_1.1 inhibition-induced overcome of DOX resistance in cancer spheroid models, we examined the effects of treatment with 10 μM PAX for 12 and 36 h on the expression levels of CYP3A4 transcripts and proteins, respectively. The PAX treatment significantly decreased the expression of CYP3A4 transcripts in the LNCaP (Figure 3A), MG-63 (Figure 3B), and SW-1353 (Figure 3C) spheroid models. Consistent with these results, a Western blot analysis showed that the expression levels of CYP3A4 proteins with a molecular weight of approximately 65 kDa (Figure 3D–F) were significantly decreased by the PAX treatment in them (*n* = 4 for each, *p* < 0.05, 0.01) (Figure 3G–I). In the human prostate cancer PC-3 spheroid model that expressed K_Ca_1.1 at a low level (less than 0.001 a.u.), the PAX treatment did not affect the expression level of CYP3A4 transcripts (*n* = 4 for each, *p* > 0.05) (Appendix A). These results strongly suggest that the K_Ca_1.1 inhibition-induced down-regulation of CYP3A4 contributed to the overcoming of DOX resistance in K_Ca_1.1-expressing cancer spheroid models.

### 2.4. Identification of CCAAT/Enhancer-Binding Protein (CEBP) Isoforms Involving K_Ca_1.1 Inhibition-Induced Down-Regulation of CYP3A4 in Cancer Spheroid Models

Martínez-Jiménez et al. (2007) reported that CCAAT/enhancer-binding protein isoforms (C/EBPα, β, and δ (CEBPA, B, and D)) are key transcriptional factors responsible for CYP3A4 expression in hepatocytes [20]. In addition, Nishizuka et al. (2014) indicated that the inhibition of the two-pore domain K^+^ channel, K_2P_10.1, reduced the expression levels of CEBPB and CEBPD in differentiated adipocytes [21]. To identify the CEBP isoform(s) contributing to the CYP3A4 transcription in the LNCaP, MG-63, and SW-1353 spheroid models, we examined the effects of the siRNA-mediated inhibition of CEBPA, CEBPB, and CEBPD on the expression level of CYP3A4 transcripts. As shown in Figure 4, the siRNA-mediated inhibition of CEBPB down-regulated CYP3A4 in LNCaP (Figure 4A), MG-63 (Figure 4B), and SW-1353 (Figure 4C) spheroid models (*n* = 4, *p* < 0.01), whereas no significant changes in CYP3A4 levels were observed following the inhibition of CEBPA and CEBPD (*n* = 4, *p* > 0.05 for each) (Figure 4A–C). Expression of CEPBP isoforms was selectively inhibited by the siRNA transfection, and the inhibition efficacy was approximately 50% for each (Appendix A). These results strongly suggest that CEBPB but not CEBPA and CEBPD regulates CYP3A4 transcription in cancer spheroid models.

We compared the expression levels of CEBPB transcripts between 2D- and 3D-cultured LNCaP, MG-63, and SW-1353 using real-time PCR. Together with CYP3A4, the expression levels of CEBPB were significantly increased in LNCaP (Figure 5A), MG-63 (Figure 5B), and SW-1353 (Figure 5C) spheroid models (*n* = 4 for each, *p* < 0.01). We then investigated the effects of K_Ca_1.1 inhibition with PAX on the expression levels of CEBPB transcripts in cancer spheroid models. As shown in Figure 5D–F, expression levels in the cancer spheroid models were significantly down-regulated by the treatment with 10 μM PAX for 12 h (*n* = 4, *p* < 0.01). These results suggest that CEBPB was responsible for the up-regulation of CYP3A4 in cancer spheroid models and indicate their potential as crucial targets for the K_Ca_1.1 inhibition-induced down-regulation of CYP3A4.

### 2.5. Involvement of the Akt-Nrf2 Signaling Pathway in CYP3A4 Transcription in Cancer Spheroid Models

Nishizuka et al. (2014) reported that the inhibition of the K_2P_10.1 K^+^ channel reduced the phosphorylation level of Akt in differentiated adipocytes [21]. Previous studies showed that the inhibition of K_Ca_1.1 also repressed the phosphorylation level of Akt in non-cancerous cells [22,23]. In the LNCaP, MG-63, and SW-1353 spheroid models, treatment with the Akt inhibitor, AZD5363 (2 μM), for 12 h significantly decreased the expression levels of CYP3A4 transcripts (*n* = 4, *p* < 0.01) (Figure 6A–C). Moreover, a co-treatment with the Akt activator, SC79 (10 μM), and PAX (10 μM) for 12 h significantly reversed the K_Ca_1.1 inhibition-induced down-regulation of CYP3A4 (*n* = 4, *p* < 0.01) (Figure 6D–F). No significant changes in the relative expression level of CYP3A4 transcripts were found by a single treatment with SC79 in LNCaP, MG-63, and SW-1353 spheroid models (*n* = 4, *p* > 0.05 vs. −/−) (Figure 6D–F).

We also examined the effects of the PAX treatment on the phosphorylation levels of Akt (Figure 7). As described above, cells were treated with PAX (10 μM) for 12 h to detect changes in the transcriptional levels of target genes; however, they were treated with PAX for 2 h to detect changes in the phosphorylation levels of Akt. The treatment with PAX for 2 h significantly reduced P-Akt/total Akt levels in the LNCaP (Figure 7A,B), MG-63 (Figure 7C,D), and SW-1353 (Figure 7E,F) spheroid models (*n* = 4, *p* < 0.01). Total Akt levels were slightly but significantly reduced (Appendix A). The inhibition of K_Ca_1.1 has been shown to induce the dephosphorylation of Akt by activating protein phosphatase 2A (PP2A) in neuroblastoma cells [22]. To elucidate the possible contribution of PP2A to the K_Ca_1.1 inhibition-induced down-regulation of CYP3A4 in cancer spheroid models, we examined the effects of the PP2A activator, FTY720-P, on the expression level of CYP3A4 transcripts. Similar to the K_Ca_1.1 inhibition-induced down-regulation of CYP3A4, the treatment with the PP2A activator, FTY720-P (5 μM), for 12 h decreased the expression levels of CYP3A4 transcripts (Figure 7G–I).

The transcriptional factor, Nrf2, has been shown to contribute to cancer stemness and chemotherapy resistance [17,24]. Active Nrf2 may be promoted by the Akt signaling pathway [25] and up-regulate CEBPB [26], suggesting that the Akt-Nrf2 signaling pathway triggers the induction of CEBPB in cancer spheroid models. CEBPB is required for Nrf2-mediated drug resistance in Nrf2-activated non-small cell lung cancer cells [26]. Treatment with the Nrf2 inhibitor, ML385 (10 μM), for 12 h significantly decreased the expression levels of CYP3A4 transcripts (*n* = 4, *p* < 0.01) (Figure 8A–C), while a co-treatment with the Nrf2 activator, NK252 (100 μM), and PAX (10 μM) for 12 h significantly reversed the K_Ca_1.1 inhibition-induced down-regulation of CYP3A4 (*n* = 4, *p* < 0.01) (Figure 8D–F). No significant changes in the relative expression level of CYP3A4 transcripts were found by a single treatment with NK252 in LNCaP, MG-63, and SW-1353 spheroid models (*n* = 4, *p* > 0.05 vs. −/−) (Figure 8D–F).

### 2.6. Involvement of the Akt-Nrf2 Signaling Pathway in CEBPB Transcription in Cancer Spheroid Models

We examined the involvement of PP2A, Akt, and Nrf2 in the expression levels of CEBPB in cancer spheroid models. As shown in Figure 9, CEBPB expression level was significantly reduced by treatment with 5 μM FTY720-P for 12 h in the LNCaP (Figure 9A), MG-63 (Figure 9B), and SW-1353 (Figure 9C) spheroid models (*n* = 4, *p* < 0.01). The co-treatment with SC79 (10 μM) and PAX (10 μM) for 12 h significantly reversed the K_Ca_1.1 inhibition-induced down-regulation of CEBPB in the LNCaP (Figure 10A), MG-63 (Figure 10B), and SW-1353 (Figure 10C) spheroid models (*n* = 4, *p* < 0.01). Moreover, the co-treatment with NK252 (100 μM) and PAX (10 μM) for 12 h significantly reversed the K_Ca_1.1 inhibition-induced down-regulation of CEBP isoforms in the LNCaP (Figure 11A), MG-63 (Figure 11B), and SW-1353 (Figure 11C) spheroid models (*n* = 4, *p* < 0.01). No significant changes in the relative expression level of CEBPB transcripts were found by a single treatment with SC79 or NK252 in LNCaP, MG-63, and SW-1353 spheroid models (*n* = 4, *p* > 0.05) (Figure 10 and Figure 11).

Therefore, in the LNCaP, MG-63, and SW-1353 spheroid models, the inactivation of the Akt-Nrf2-CEBPB axis through the K_Ca_1.1 inhibition-induced activation of PP2A may have contributed to the transcriptional repression of CYP3A4, triggering overcome of DOX resistance.

### 2.7. No Involvement of the ERK and JNK Signaling Pathways in the K_Ca_1.1 Inhibition-Induced Down-Regulation of CEBPB and CYP3A4 in Cancer Spheroid Models

In addition to the Akt-Nrf2 signaling pathway, the JNK/Nrf2 and ERK/Nrf2 pathways contribute to the acquisition of DOX resistance [17,27] and the regulation of CYP3A4 expression [28,29]. A recent study showed that the Ca^2+^-activated K^+^ channel, K_Ca_3.1, potentially regulates the JNK and ERK signaling pathways [30]. We, therefore, examined the involvement of JNK and ERK in the expression levels of CEBPB and CYP3A4 in cancer spheroid models. As shown in Figure 12, the decrease in CEBPB expression level by K_Ca_1.1 inhibition was not affected by the co-treatment with a JNK activator, anisomycin (1 μM, ANM), or an ERK activator, senkyunolide I (1 μM, SKL-I), for 12 h in the LNCaP (Figure 12A), MG-63 (Figure 12B), and SW-1353 (Figure 12C) spheroid models (*n* = 4, *p* > 0.05). Consistent with these results, the decrease in CYP3A4 expression level by K_Ca_1.1 inhibition was not affected by the co-treatment with ANM or SKL-I for 12 h in them (*n* = 4, *p* > 0.05) (Figure 12D–F). No significant changes in the relative expression level of CEBPB and CYP3A4 transcripts were found by a single treatment with ANM or SKL-I in LNCaP, MG-63, and SW-1353 spheroid models (*n* = 4, *p* > 0.05) (Figure 12). These results suggest that the JNK and ERK signaling pathways may not contribute to the K_Ca_1.1 inhibition-induced down-regulation of CEBPB and CYP3A4 in the cancer spheroid models examined. 

## 3. Discussion

Previous studies demonstrated that the acquisition of chemoresistance by spheroid formation in K_Ca_1.1-expressing LNCaP, MG-63, and SW-1353 cells was caused by the increased expression of multidrug resistance-related proteins (MRP1/ABCC1 and MRP5/ABCC5), and the inhibition of K_Ca_1.1 overcame this chemoresistance by down-regulating MRPs [4,5]. Recent studies reported that the higher intratumoral expression of drug-metabolizing CYPs was associated with resistance to chemotherapies [31,32]. In the present study, we elucidated the involvement of CYPs in the mechanisms underlying the overcoming of DOX resistance through the inhibition of K_Ca_1.1 in cancer spheroid models with CSC phenotypes. The main results obtained were as follows. (1) Among the CYP isoforms involved in DOX metabolism, the expression levels of CYP3A4 and CYP2B6 were markedly increased by 3D spheroid formation in LNCaP, MG-63, and SW-1353 cells (Figure 1). (2) DOX resistance was overcome by the siRNA-mediated inhibition of CYP3A4 but not CYP2B6 (Figure 2). (3) The pharmacological inhibition of K_Ca_1.1 down-regulated CYP3A4 expression (Figure 3). (4) The K_Ca_1.1 inhibition-induced down-regulation of CYP3A4 was mediated by the Akt-Nrf2-CEBPB axis (Figure 4, Figure 5, Figure 6, Figure 7, Figure 8, Figure 9, Figure 10 and Figure 11). Litviakov reported that stemness genes are located on 11.23, 21.13, 31.2, and 32.1 of 7q; however, the chromosomal location of CYP3A4 (7q22.1) is different from them [33]. The present results, in addition to previous findings [4,5], indicated that 3D cancer spheroids as in vitro models to mimic the TME of solid cancers are a valid model for investigating resistance to anti-cancer drugs through the induction of drug efflux transporter MRPs and drug-metabolizing CYPs (Figure 13).

The activation of the PI3K/Akt signaling pathway is important for promoting cancer stemness and resistance to anti-cancer therapies [34,35]. Furthermore, PP2A is necessary for inactivating the Akt signaling pathway through the dephosphorylation of P-Akt [36,37]. Therefore, the activation of PP2A has potential as a therapeutic strategy to overcome resistance to anti-cancer therapies. Eil et al. (2016) indicated that an increase in the intracellular concentration of K^+^ ([K^+^]_i_) followed by that in the extracellular concentration of K^+^ ([K^+^]_e_) in the TME extracellular fluid activated PP2A activity in CD8^+^ T cells, resulting in the down-regulated expression of pro-inflammatory cytokines through the inactivation of the Akt signaling pathway [38]. In addition, Maqoud et al. (2018) indicated that the pharmacological inhibition of K_Ca_1.1 induced the dephosphorylation of Akt by activating protein phosphatases, such as PP2A, in neuroblastoma cells [22]. As shown in Figure 7A–F, phosphorylated Akt levels were reduced by the inhibition of K_Ca_1.1 in cancer spheroid models. The activation of PP2A by FTY720-P reduced the expression levels of the CYP3A4 (Figure 7G–I) and CEBPB (Figure 9) in cancer spheroid models. These results suggest that the intracellular accumulation of K^+^ through the inhibition of K_Ca_1.1 may activate PP2A, resulting in the down-regulation of CYP3A4 via the inactivation of the Akt signaling pathway. Plasma membrane depolarization by the inhibition of K_Ca_1.1 may also be involved in the inactivation of the Akt signaling pathway through a decrease in the intracellular concentration of Ca^2+^. In addition to the Akt signaling pathway, the JNK and ERK signaling pathways are also important for the acquisition of DOX resistance and the transcriptional regulation of CYP3A4 in cancerous and non-cancerous cells [17,27,28,29]. In the previous study, we showed the involvement of K_Ca_3.1 in the regulation of JNK and ERK signaling pathways in macrophages [30]. However, the significant reverse of the K_Ca_1.1 inhibition-induced down-regulation of CEBPB and CYP3A4 by K_Ca_1.1 inhibition was not observed following the treatment with JNK and ERK inhibitors in cancer spheroid models (Figure 12). 

Nrf2 plays an emerging role in the maintenance of cancer stemness and the acquisition of chemoresistance [24,39]. In the Keap-Nrf2 system, Keap1 functions as a negative regulator of Nrf2 by inhibiting the cytoplasmic-to-nuclear translocation of Nrf2. Previous studies reported the regulation of chemoresistance through the Akt/Nrf2 signaling pathway [25,39]. The activation of Nrf2 by the knockdown of Keap1 increased CEBPB expression in adipocytes [26,40]. In addition, CEBP isoforms contributed to the basal expression of CYP genes, including CYP3A4, in cancerous and non-cancerous cells [41,42]. CEBPB and Nrf2 cooperatively contributed to anti-cancer drug resistance [26,41]. In the present study, siRNA mediated the inhibition of CEBPB but not CEBPA, CEBPD significantly decreased the expression levels of CYP3A4 transcripts (Figure 4), CEBPB was markedly up-regulated by the spheroid formation in LNCaP, MG-63, and SW-1353 cells, and the inhibition of K_Ca_1.1 mostly reversed its expression (Figure 5). Moreover, Akt and Nrf2 activators both significantly reversed the K_Ca_1.1 inhibition-induced down-regulation of CYP3A4 (Figure 6D–F and Figure 8D–F) and CEBPB (Figure 10 and Figure 11). These results suggest that the K_Ca_1.1 inhibition-induced down-regulation of CYP3A4 was mediated by the Akt-Nrf2-CEBPB axis. The CYP3A4 and CEBPB expressions were not increased by the single treatment with Akt and Nrf2 activators, suggesting that the Akt-Nrf2 signaling pathway is almost sufficiently activated by the 3D spheroid formation. As shown in Figure 12, the JNK and ERK signaling pathways did not affect the K_Ca_1.1 inhibition-induced down-regulation of CEBPB and CYP3A4 in the cancer spheroid models examined; however, the JNK inhibition with SP600125 (1 μM) significantly reduced the expression levels of CEBPB and CYP3A4, and co-treatment with NK252 reversed them (Appendix A). These results suggest that the JNK/Nrf2 signaling pathway may be somewhat responsible for the acquisition of DOX resistance in cancer spheroid models in a K_Ca_1.1-independent manner. The possible involvement of the ERK signaling pathway in DOX resistance was not observed in the cancer spheroid models examined (Appendix A).

In previous studies, we concluded that (1) increases in MRP1 by spheroid formation in MG-63 and SW-1353 cells were reduced by the inhibition of K_Ca_1.1 through the Nrf2 signaling pathway [5], and (2) increases in MRP5 by spheroid formation in LNCaP cells were reduced by the inhibition of K_Ca_1.1 through Nanog signaling [4]. Nanog, a component of pluripotent stem cell markers, leads to resistance to anti-cancer drugs by the up-regulation of drug efflux transporters, such as MDR1 and MRP1 [43]. The human gene database, GeneCards^®^ (https://www.genecards.org) accessed on 17 May 2023, also represents Nanog-mediated MRP1 expression. A recent study showed that Nanog was regulated by the PI3K/Akt signaling pathway [19]. Taken together with the present results, MRP1 and MRP5 may both be up-regulated through the activation of the Akt-Nrf2-Nanog axis by cancer spheroid formation in LNCaP, MG-63, and SW-1353 cells (Figure 12). The findings of previous studies strongly support this assumption. Tazari et al. (2007) identified MRP1 as a downstream target for the PI3K/Akt signaling pathway in cancer cells [44]. Yu et al. (2021) showed that the PI3K/Akt/MRP1 cascade was responsible for the acquisition of DOX resistance [45]. In addition, Gao et al. (2013) reported that MRP5 was a downstream target for the PI3K/Akt/Nrf2 signaling pathway [46]. In the LNCaP, MG-63, and SW-1353 spheroid models in the present study, the expression level of Nanog transcripts was significantly reduced by the inhibition of K_Ca_1.1, Akt, and Nrf2 and the activation of PP2A for 12 h (Appendix A). 

Taxanes, such as paclitaxel (PTX) and docetaxel (DTX), are predominantly metabolized in the liver by CYP isoforms (i.e., CYP3A4 and CYP2C8) [47]. Investigations have recently focused on intratumoral CYP isoforms, which are associated with resistance to chemotherapies [31,32]. A previous study showed that resistance to taxanes acquired by 3D spheroid formation in MG-63 and SW-1353 was overcome by the inhibition of K_Ca_1.1 [5]. However, the siRNA-mediated inhibition of CYP3A4 did not reverse resistance to taxanes (Appendix A). Van Eijk et al. (2019) indicated that taxanes increased the expression of CYP3A4 in cancer cells [32]. In the present study, the K_Ca_1.1 inhibition-induced down-regulation of CYP3A4 was mostly reversed by the co-treatment with taxanes (*p* < 0.01 vs. vehicle control) (Appendix A). Therefore, CYP3A4 is not a major contributor to resistance to taxanes in sarcoma spheroid models.

Plasma membrane solute carrier influx (SLC) transporters play an essential role in the cellular uptake of chemotherapy drugs, and several SLC members are responsible for the uptake of DOX [48]. Among the five SLC isoforms (SLC21A3, 21A8, 22A4, 22A7, and 22A16) that may be involved in the influx of DOX into cells, SLC22A16 was markedly down-regulated by spheroid formation in MG-63 and SW-1353 cells (Appendix A). On the other hand, it was up-regulated in the LNCaP spheroid model (Appendix A). These results suggest that SLC22A16 contributed to the acquisition of DOX resistance in the MG-63 and SW-1353 spheroid models. However, no significant changes in the expression levels of SLC22A16 transcripts were observed by the treatment with 10 μM PAX for 12 h (Appendix A), suggesting that SLC22A16 is not related to the mechanisms underlying the K_Ca_1.1 inhibition-induced overcoming DOX resistance.

## 4. Materials and Methods

### 4.1. Chemicals and Reagents

The following chemicals and reagents were used: PAX, AZD5363, anisomycin, PD169316, and FTY720 phosphate (FTY720-P) from Cayman Chemical (Ann Arbor, MI, USA); PTX, DTX, DOX, and culture media (RPMI 1640 and DMEM) from FUJIFILM Wako Pure Chemical (Tokyo, Japan); fetal bovine serum (FBS) from Sigma-Aldrich (St. Louis, MO, USA); Silencer^®^ Select Pre-designed siRNAs as a negative control No. 1, human CYP3A4 (siRNA ID #: s3846), human CYP2B6 (s3819), human CEBPA (s2888), human CEBPB (s2892), and CEBPD (s2895) from Life Technologies Japan (Tokyo, Japan); anti-phospho-Akt (Ser473) polyclonal (rabbit) and anti-Akt monoclonal (mouse) antibodies from BioLegend (San Diego, CA, USA); an anti-ACTB monoclonal (mouse) antibody from Medical & Biological Laboratories (Nagoya, Japan); an anti-CYP3A4 polyclonal (rabbit) (A2544) antibody from ABclonal (Tokyo, Japan); anti-rabbit and mouse horseradish peroxidase-conjugated IgG antibodies from Merck Millipore (Darmstadt, Germany); ketoconazole, SC79, NK252, stattic, SCH772984, and senkyunolide I from MedChemExpress (Monmouth Junction, NJ, USA); WST-1 from Dojindo (Kumamoto, Japan), ReverTra Ace from ToYoBo (Osaka, Japan); SP600125 from LC Laboratories (Woburn, MA, USA); flat-bottomed dishes and plates from Corning (Corning, NY, USA); PrimeSurface 96U plates from Sumitomo Bakelite (Tokyo, Japan); Luna Universal qPCR Master Mix from New England Biolabs Japan (Tokyo, Japan); Lipofectamine RNAiMAX Transfection Reagent from Thermo Fisher Scientific (Waltham, MA, USA); and ML385 from Selleckchem (Huston, TX, USA). The other chemicals used in the present study were from Sigma-Aldrich, FUJIFILM Wako Pure Chemical, and Nacalai Tesque (Kyoto, Japan), unless otherwise stated.

### 4.2. Cell Culture

The human osteosarcoma cell line, MG-63, chondrosarcoma cell line, SW-1353, and prostate cancer cell lines, LNCaP and PC-3, were purchased from the RIKEN Cell Bank (Osaka, Japan). Sarcoma cells (MG-63 and SW-1353) and prostate cancer cells (LNCaP and PC-3) were cultured in DMEM and RPMI 1640 media, respectively, supplemented with 10% FBS and penicillin (100 units/mL)-streptomycin (100 μg/mL) [4,5]. All cell lines were cultured in a humidified atmosphere containing 5% CO_2_ at 37 °C. Flat-bottomed dishes and plates were used in the 2D cell culture. PrimeSurface 96 U plates were used for 3D spheroid formation. Cell suspensions of MG-63, SW-1353, and LNCaP were seeded onto PrimeSurface 96U plates at 10^4^ cells/well and then cultured for 5 (MG-63 and SW-1353) and 7 (LNCaP) days.

### 4.3. RNA Extraction, cDNA Synthesis, and Real-Time PCR

Total RNA was isolated from cells by the conventional acid guanidium thiocyanate–phenol–chloroform method. Reverse transcription was performed using ReverTra Ace with random hexanucleotides. cDNAs were prepared from cells that were individually seeded on another day. Quantitative real-time PCR was conducted using the Luna Universal qPCR Master Mix and the Applied Biosystems 7500 Fast Real-Time PCR system (Thermo Fisher Scientific) [4,5]. The following PCR primers of human origin were used: CYP1B1 (NM_000104, 1243-1362, 1200 bp); CYP2A6 (NM_000762, 254-373, 120 bp); CYP2A7 (NM_000764, 473-592, 120 bp); CYP3A4 (NM_017460, 961-1080, 120 bp); CYP3A5 (NM_000777, 869-987, 119 bp); CYP2B6 (NM_000767, 1031-1150, 120 bp); CYP2C8 (NM_000770, 189-308, 120 bp); CYP2D6 (NM_000106, 639-758, 120 bp); CYP4Z1 (NM_178134, 502-621, 120 bp); CCAAT/enhancer binding protein (C/EBP) α (CEBPA, NM_001287435, 1395-1515, 121 bp); C/EBP β (CEBPB, NM_001285878, 1216-1310, 107 bp); C/EBP δ (CEBPD, NM_005195, 1094-1193, 120 bp); SLC21A3 (NM_134431, 2175-2294, 120 bp); SLC21A8 (NM_019844, 893-1012, 120 bp); SLC22A4 (NM_003059, 1018-1135, 118 bp); SLC22A7 (NM_006672, 439-558, 120 bp); SLC22A16 (NM_033125, 875-994, 120 bp); Nanog (NM_024865, 382-501, 120 bp); and ACTB (NM_001101, 411-511, 101 bp). No significant differences in the cycle threshold (Ct) values were not found between the 2D- and 3D-cultured cells examined. Relative expression levels were calculated using the 2^−ΔΔCt^ method and normalized to ACTB.

### 4.4. Western Blots

Whole-cell lysates were prepared from cells that were individually seeded on another day and were extracted using a RIPA buffer. Equal amounts of protein were subjected to SDS-PAGE and immunoblotting with anti-CYP3A4 polyclonal (rabbit) (1:1000) (approximately 65 kDa), anti-Phospho-Akt (P-Akt) polyclonal (rabbit) (1:1000) (approximately 60 kDa), anti-Akt monoclonal (mouse) (1:1000) (approximately 60 kDa), and anti-ACTB monoclonal (mouse) (1:15,000) (approximately 45 kDa) antibodies (at 4 °C, overnight), and were then incubated with anti-rabbit (1:7500) or -mouse (1:15,000) IgG horseradish peroxidase-conjugated antibodies. An ECL Western Blotting Detection Reagent was used to detect the bound antibody. The resulting images were analyzed using an Amersham Imager 600 (GE Healthcare, Tokyo, Japan). In Figure 3, the optical density of the protein band signal relative to that of the ACTB signal was calculated using ImageJ software (Ver. 1.42, NIH, USA). In Figure 7, the ratio of the P-Akt signal to total Akt was calculated by quantifying the P-Akt normalized to total Akt. Protein expression levels in the vehicle control were then expressed as 1.0.

### 4.5. Cell Viability Assay

The WST-1 assay is used to assess the in vitro cytotoxicity of chemotherapy [4,5]. Briefly, cells were cultured at a density of 10^4^ cells/mL in duplicate in 96-well plates for 5 or 7 days (for the 3D culture) and 1 day (for the 2D culture). Cells were then treated with the drugs used for chemotherapy (PTX, DTX, and DOX) for 2 days. Two hours after the addition of the WST-1 reagent to each well, absorbance was measured using a microplate reader, SpectraMax 384 (Molecular Devices Japan, Tokyo, Japan), at a test wavelength of 450 nm and a reference wavelength of 650 nm.

### 4.6. siRNA Transfection

A lipofectamine RNAiMAX reagent (Thermo Fisher Scientific) was used in the siRNA-mediated inhibition of CYP and CEBP isoforms, according to the manufacturer’s protocol. Silencer Select Pre-designed siRNAs for the negative control (siCont), human CYP2B6 (siCYP2B6), human CYP3A4 (siCYP3A4), human CEBPA (siCEBPA), human CEBPB (siCEBPB), and human CEBPD (siCEBPD) were transfected into adherent monolayer cells. Twenty-four hours later, transfected cells were seeded onto PrimeSurface 96U plates. The expression levels of the target transcripts were assessed using real-time PCR examinations.

### 4.7. Statistical Analysis

Statistical analyses were performed using XLSTAT (version 2013.1). To assess the significance of differences between two groups and among multiple groups, unpaired/paired Student’s *t*-tests with Welch’s correction or Tukey’s test were used. Results with a *p*-value of <0.05 were considered to be significant. Data are presented as means ± SEM.

## 5. Conclusions

Previous studies indicated that the inhibition of K_Ca_1.1 may be developed as a novel method to recover DOX sensitivity in K_Ca_1.1-overexpressing and DOX-resistant CSCs. The present study is the first to support the notion that the inhibition of K_Ca_1.1 may manipulate CYP3A4, which is involved in the metabolism of many anti-cancer drugs and revealed the K_Ca_1.1 inhibition-mediated overcoming DOX resistance in K_Ca_1.1 overexpressing cancer spheroid models. The Akt/Nrf2/CEBPB axis was mainly involved in the induction of CYP3A4 by cancer spheroid formation, and the inactivation of Akt through the inhibition of K_Ca_1.1 induced the down-regulation of CYP3A4, which is involved in DOX metabolism. The results obtained herein provide a valuable in vitro background for future in vivo studies on intratumoral CYP-mediated drug resistance.

## Figures and Tables

**Figure 1 ijms-24-15672-f001:**
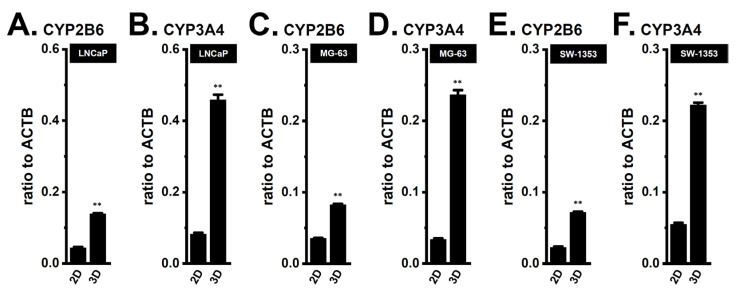
Comparison of CYP2B6 and CYP3A4 mRNA expression levels between 2D monolayers and 3D spheroid models of LNCaP, MG-63, and SW-1353 cells. (**A**–**F**) Real-time PCR examination of CYP2B6 (**A**,**C**,**E**) and CYP3A4 (**B**,**D**,**F**) in 2D monolayers and 3D spheroid models of LNCaP (**A**,**B**), MG-63 (**C**,**D**), and SW-1353 (**E**,**F**) cells (*n* = 4 for each). Expression levels are shown as a ratio to ACTB. **: *p* < 0.01 vs. ‘2D’.

**Figure 2 ijms-24-15672-f002:**
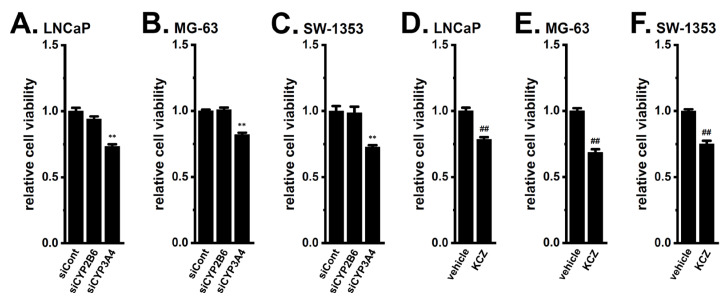
Effects of the siRNA-mediated inhibition of CYP2B6 and CYP3A4 and treatment with the potent CYP3A4 inhibitor, ketoconazole, on DOX resistance acquired by 3D spheroid models of LNCaP, MG-63, and SW-1353 cells. (**A**–**C**) Effects of treatment with 1 μM DOX for 48 h on the cell viability of cancer spheroid models transfected with negative control siRNA (siCont), human CYP2B6 siRNA (siCYP2B6), and human CYP3A4 siRNA (siCYP3A4) using the WST-1 assay (*n* = 5 for each). (**D**–**F**) Effects of a co-treatment with 1 μM DOX and vehicle (0.1% DMSO) or 1 μM ketoconazole (KCZ) for 48 h on the cell viability of cancer spheroid models (*n* = 5 for each). Cell viabilities in the siCont-transfected (**A**–**C**) and vehicle-treated (**D**–**F**) groups were expressed as 1.0. **: *p* < 0.01 vs. siCont; ^##^: *p* < 0.01 vs. the vehicle control.

**Figure 3 ijms-24-15672-f003:**
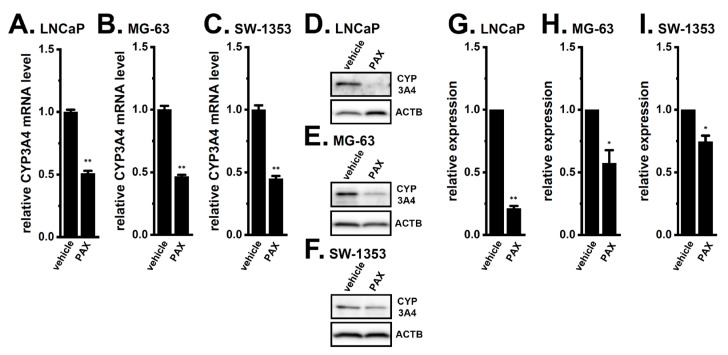
Effects of K_Ca_1.1 inhibition on CYP3A4 expression in 3D spheroid models of LNCaP, MG-63, and SW-1353 cells. (**A**–**C**) Real-time PCR examination of CYP3A4 in cancer spheroid models treated with the vehicle and 10 μM PAX for 12 h. After normalization to ACTB mRNA expression levels, CYP3A4 mRNA expression levels in the vehicle control (vehicle) were expressed as 1.0 (*n* = 4 for each). (**D**–**F**) Protein expressions of CYP3A4 in protein lysates of cancer spheroid models treated with the vehicle and 10 μM PAX for 36 h. Blots were probed with anti-CYP3A4 (approximately 65 kDa) and anti-ACTB (approximately 45 kDa) antibodies. (**G**–**I**) Summarized results were obtained as the optical densities of CYP3A4 from (**D**–**F**). After normalization of the optical densities of protein band signals with the ACTB signal, the optical density in the vehicle-treated group (vehicle) was expressed as 1.0. *, **: *p* < 0.05, 0.01 vs. the vehicle control.

**Figure 4 ijms-24-15672-f004:**
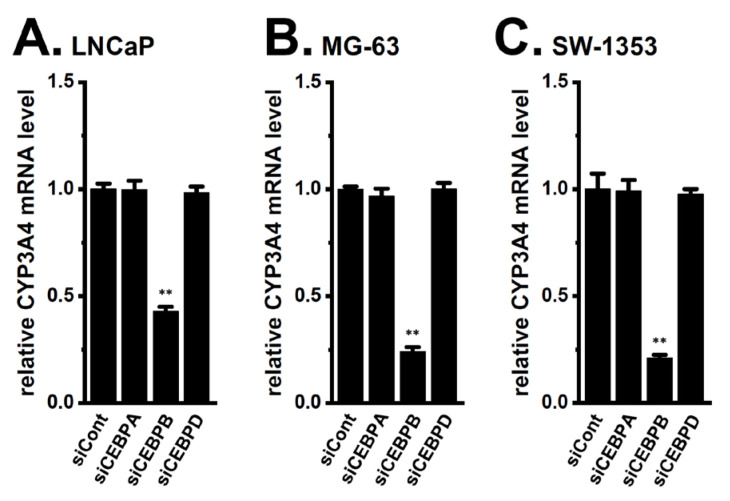
Effects of the siRNA-mediated inhibition of CEBPA, CEBPB, and CEBPD on the expression level of CYP3A4 transcripts in 3D spheroid models of LNCaP, MG-63, and SW-1353 cells. (**A**–**C**) Real-time PCR examination of CYP3A4 in cancer spheroid models transfected with negative control siRNA (siCont), human CEBPA siRNA (siCEBPA), and human CEBPB siRNA (siCEBPB), human CEBPD siRNA (siCEBPD). After normalization to ACTB mRNA expression levels, the CYP3A4 mRNA expression levels in the siCont-transfected group were expressed as 1.0. **: *p* < 0.01 vs. siCont.

**Figure 5 ijms-24-15672-f005:**
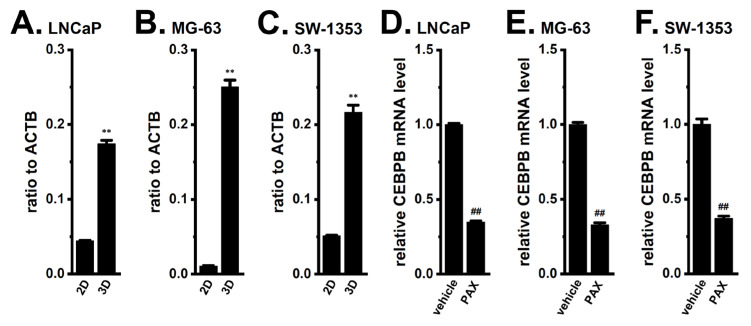
Comparison of CEBPB mRNA expression levels between 2D monolayers and 3D spheroid models of LNCaP, MG-63, and SW-1353 cells and effects of the K_Ca_1.1 inhibition on CEBPB mRNA expression levels. (**A**–**C**) Real-time PCR examination of CEBPB in 2D monolayers and 3D spheroid models of cancer cells (*n* = 4 for each). Expression levels are shown as a ratio to ACTB. (**D**–**F**) Real-time PCR examination of CEBPB in vehicle- and PAX (10 μM)-treated cancer spheroid models of cancer cells for 12 h (*n* = 4 for each). After normalization to ACTB mRNA expression levels, CEBPB mRNA expression levels in the vehicle control were expressed as 1.0. **: *p* < 0.01 vs. ‘2D’; ^##^: *p* < 0.01 vs. the vehicle control.

**Figure 6 ijms-24-15672-f006:**
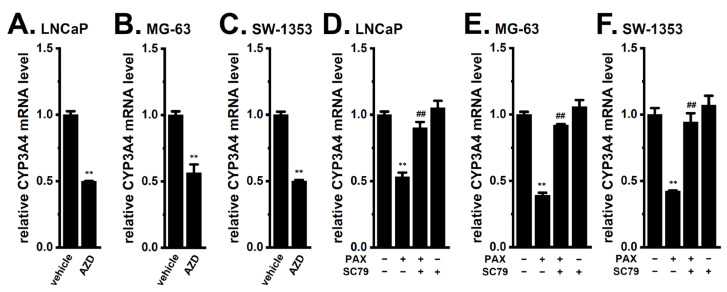
Effects of Akt inhibition on CYP3A4 mRNA expression levels and effects of Akt activation on the K_Ca_1.1 inhibition-induced down-regulation of CYP3A4 in 3D spheroid models of LNCaP, MG-63, and SW-1353 cells. (**A**–**C**) Real-time PCR examination of CYP3A4 in cancer spheroid models treated with the vehicle and 2 μM AZD5363 (AZD) for 12 h (*n* = 4 for each). (**D**–**F**) Real-time PCR examination of CYP3A4 in cancer spheroid models treated (+) or untreated (−) with 10 μM PAX and 10 μM SC79 for 12 h (*n* = 4 for each). After normalization to ACTB mRNA expression levels, CYP3A4 mRNA expression levels in the vehicle control (−/−) were expressed as 1.0. **: *p* < 0.01 vs. the vehicle control (**A**–**C**) and −/− (**D**–**F**); ^##^: *p* < 0.01 vs. +/−.

**Figure 7 ijms-24-15672-f007:**
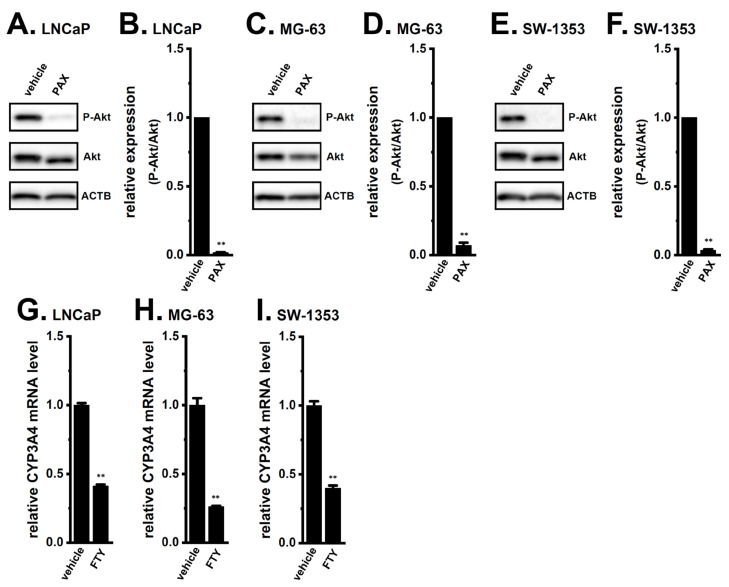
Effects of K_Ca_1.1 inhibition on expression levels of phosphorylated Akt (P-Akt) and total Akt (Akt) proteins and effects of PP2A activation on expression levels of CYP3A4 transcripts in 3D spheroid models of LNCaP, MG-63, and SW-1353 cells. (**A**,**C**,**E**) Protein lysates of cancer spheroid models treated with the vehicle or 10 μM PAX for 2 h were probed by immunoblotting with anti-P-Akt, anti-Akt, and anti-ACTB antibodies. (**B**,**D**,**F**) The ratio of the P-Akt signal was calculated by quantifying the P-Akt normalized to Akt, and the vehicle-treated group (vehicle) was expressed as 1.0 (*n* = 4 for each). (**G**–**I**) Real-time PCR examination of CYP3A4 in cancer spheroid models treated with the vehicle and FTY720-P (FTY, 5 μM) for 12 h (*n* = 4 for each). After normalization to ACTB mRNA expression levels, CYP3A4 mRNA expression levels in the vehicle control (vehicle) were expressed as 1.0. **: *p* < 0.01 vs. the vehicle control.

**Figure 8 ijms-24-15672-f008:**
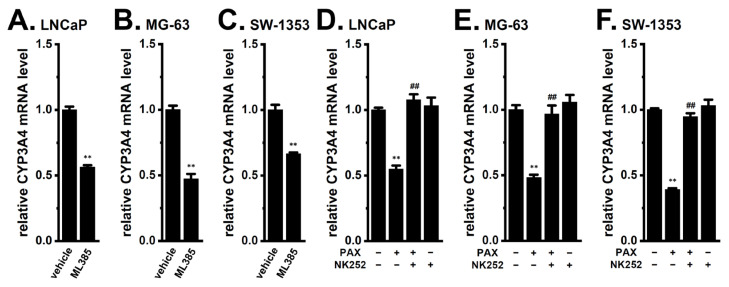
Effects of Nrf2 inhibition on CYP3A4 mRNA expression levels and effects of Nrf2 activation on the K_Ca_1.1 inhibition-induced down-regulation of CYP3A4 in 3D spheroid models of LNCaP, MG-63, and SW-1353 cells. (**A**–**C**) Real-time PCR examination of CYP3A4 in cancer spheroid models treated with the vehicle and 10 μM ML385 for 12 h (*n* = 4 for each). (**D**–**F**) Real-time PCR examination of CYP3A4 in cancer spheroid models treated (+) or untreated (−) with 10 μM PAX and 100 μM NK252 for 12 h (*n* = 4 for each). After normalization to ACTB mRNA expression levels, CYP3A4 mRNA expression levels in the vehicle control (−/−) were expressed as 1.0. **: *p* < 0.01 vs. the vehicle control (**A**–**C**) and −/− (**D**–**F**); ^##^: *p* < 0.01 vs. +/−.

**Figure 9 ijms-24-15672-f009:**
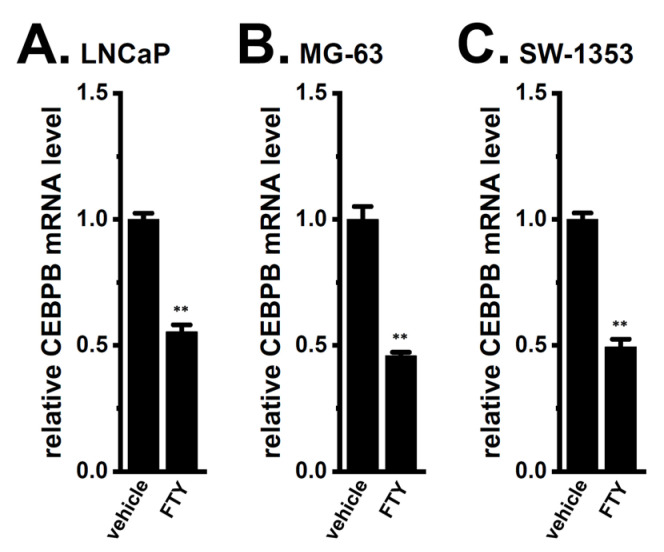
Effects of the activation of PP2A on CEBPB mRNA levels in 3D spheroid models of LNCaP, MG-63, and SW-1353 cells. (**A**–**C**) Real-time PCR examination of CEBPB transcripts in vehicle- and 5 μM FTY720-P (FTY)-treated cancer spheroid models for 12 h (*n* = 4 for each). After normalization to ACTB mRNA expression levels, CEBPB mRNA expression levels in the vehicle control were expressed as 1.0. **: *p* < 0.01 vs. the vehicle control.

**Figure 10 ijms-24-15672-f010:**
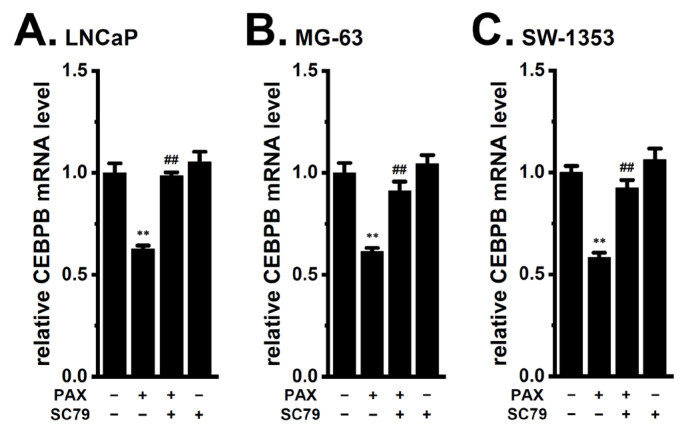
Effects of Akt activation on the K_Ca_1.1 inhibition-induced down-regulation of CEBPB in 3D spheroid models of LNCaP, MG-63, and SW-1353 cells. (**A**–**C**) Real-time PCR examination of CEBPB in cancer spheroid models treated (+) or untreated (−) with 10 μM PAX and 10 μM SC79 for 12 h (*n* = 4 for each). After normalization to ACTB mRNA expression levels, CEBPB mRNA expression levels in the vehicle control (−/−) were expressed as 1.0. **: *p* < 0.01 vs. −/−; ^##^: *p* < 0.01 vs. +/−.

**Figure 11 ijms-24-15672-f011:**
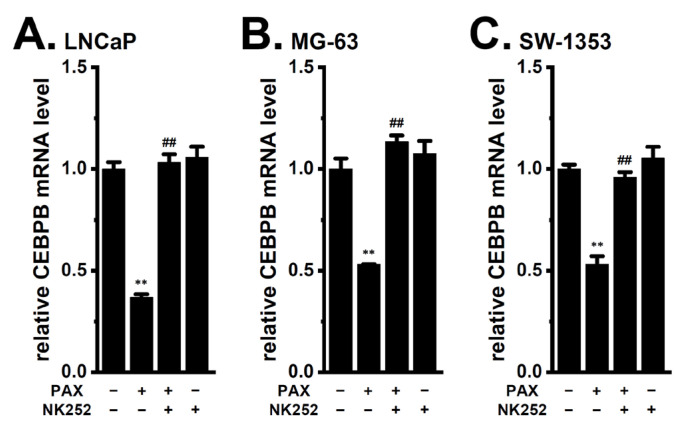
Effects of Nrf2 activation on the K_Ca_1.1 inhibition-induced down-regulation of CEBPB in 3D spheroid models of LNCaP, MG-63, and SW-1353 cells. (**A**–**C**) Real-time PCR examination of CEBPB in cancer spheroid models treated (+) or untreated (−) with 10 μM PAX and 100 μM NK252 for 12 h (*n* = 4 for each). After normalization to ACTB mRNA expression levels, CEBPB mRNA expression levels in the vehicle control (−/−) were expressed as 1.0. **: *p* < 0.01 vs. −/−; ^##^: *p* < 0.01 vs. +/−.

**Figure 12 ijms-24-15672-f012:**
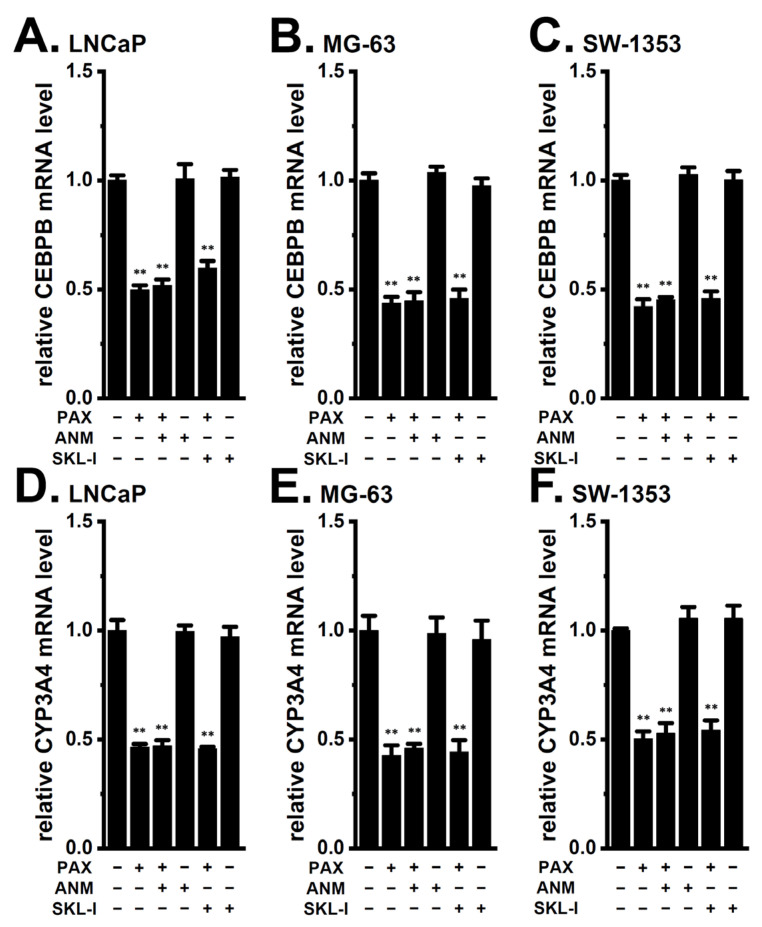
Effects of JNK and ERK activation on the K_Ca_1.1 inhibition-induced down-regulation of CEBPB and CYP3A4 in 3D spheroid models of LNCaP, MG-63, and SW-1353 cells. (**A**–**F**) Real-time PCR examination of CEBPB and CYP3A4 in cancer spheroid models treated (+) or untreated (−) with 10 μM PAX, 1 μM anisomycin (ANM), and 1 μM senkyunolide I (SKL-I) for 12 h (*n* = 4 for each). After normalization to ACTB mRNA expression levels, CEBPB and CYP3A4 mRNA expression levels in the vehicle control (−/−/−) were expressed as 1.0. **: *p* < 0.01 vs. −/−/−.

**Figure 13 ijms-24-15672-f013:**
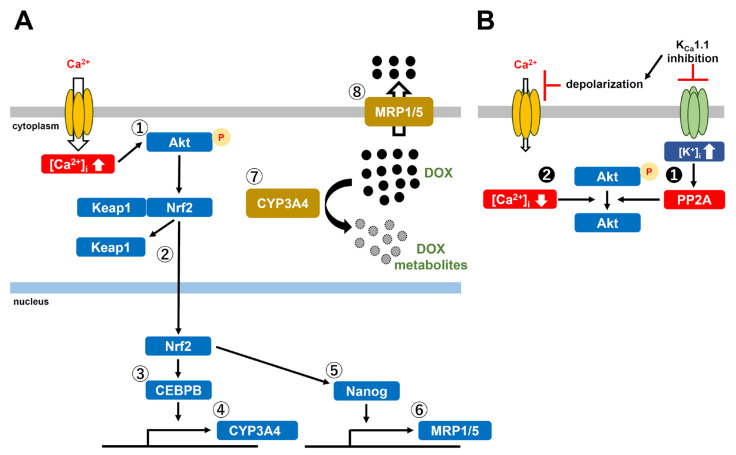
Schematic cellular representation of the K_Ca_1.1 inhibition-induced overcoming DOX resistance in K_Ca_1.1-expressing cancer spheroid models. (**A**) Acquisition of DOX resistance through the Akt-Nrf2-CEBPB axis in 3D spheroid models. The Akt activation (➀) by spheroid formation-induced phosphorylation increases the nuclear translocation of Nrf2 after release from Keap1 (➁). The activated Nrf2 increases the CEBPB transcription (➂) and further up-regulates CYP3A4 (➃). Also, the activated Nrf2 increases the Nanog transcription (➄) and further up-regulates MRP1/5 (➅). Increased CYP3A4 (➆) and MRP1/5 (➇) promote DOX metabolism and efflux. (**B**) Inactivation of the Akt by the K_Ca_1.1 inhibition-induced activation of PP2A (❶) and subsequent dephosphorylation of Akt. Decrease in [Ca^2+^]_i_ by the K_Ca_1.1 inhibition-induced plasma membrane depolarization (❷) and dephosphorylation of Akt. ●: DOX, ●: DOX metabolites.

## Data Availability

Not applicable.

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
