# Peer review of "Down-Regulation of CYP3A4 by the KCa1.1 Inhibition Is Responsible for Overcoming Resistance to Doxorubicin in Cancer Spheroid Models"

_ijms, 2023, doi:10.3390/ijms242115672_

Round 1
Reviewer 1 Report (Previous Reviewer 2)
Comments and Suggestions for Authors
my comments have been well-addressed, i have no further questions
Author Response
We really appreciate the reviewer's valuable and adequate comments and careful reading.
Reviewer 2 Report (New Reviewer)
Comments and Suggestions for Authors
The manuscript describes the role of CYP3A4 in the resistance to doxorubicin in different 3D cell culture models. The findings point to a regulatory role of KCa1.1. on CYP3A4 expression and hence, in drug resistance. Regulation of CYP3A4 takes place via Akt, Nrf2 and CEBPB. The authors confirm the key findings using three different cancer cell lines. The experimental design is correct and the conclusions are supported by the data. The following aspects should be addressed before publication:
1. Lines 50-53. The sentence “Previous studies reported a novel regulatory mechanism for KCa1.1 protein expression and channel activity, mediating the E3 ubiquitin ligase, FBXW7, in 3D spheroid models of osteosarcoma MG-63, chondrosarcoma SW-1353, and prostate cancer LNCaP cells [4-6]” is not clear. Please, check and reformulate.
2. Line 78. The reference to the inhibition of CYP3A4 by ketoconazole should be corrected. Actually, reference 16 (Greenblatt et al.) seems more appropriate that reference 15 (Christowitz et al.) as indicated in the manuscript.
3. Figure 5. Did the authors investigate the stability of ACTB and, therefore, its suitability as housekeeping gene between 2D and 3D cultures?
4. Line 821. The sentence needs to be reformulated. Cells (not PAX) were treated.
5. Figure 8. Considering the regulatory role of Nrf2 activation on CYP3A4 expression, it could be expected that treatment with NK252 alone does increase CYP3A4 expression. However, this was not the case. Please, discuss.
Comments on the Quality of English LanguageMinor edition of English required.
Author Response
We would like to thank the reviewer for his/her valuable comments. We have attended to all the points raised by the reviewers. Each comment is highlighted below with our response underneath.
- Lines 50-53. The sentence “Previous studies reported a novel regulatory mechanism for KCa1.1 protein expression and channel activity, mediating the E3 ubiquitin ligase, FBXW7, in 3D spheroid models of osteosarcoma MG-63, chondrosarcoma SW-1353, and prostate cancer LNCaP cells [4-6]” is not clear. Please, check and reformulate.
>We appreciate the reviewer’s careful reading of our manuscript. We amended this sentence (Page 2, Lines 44-47 in the revised version).
- Line 78. The reference to the inhibition of CYP3A4 by ketoconazole should be corrected. Actually, reference 16 (Greenblatt et al.) seems more appropriate than reference 15 (Christowitz et al.) as indicated in the manuscript.
> We really appreciate the reviewer’s careful reading of our manuscript. We switched Ref. 15 and 16.
- Figure 5. Did the authors investigate the stability of ACTB and, therefore, its suitability as housekeeping gene between 2D and 3D cultures?
>As pointed out by the reviewer, it is important to confirm expression levels of ACTB between 2D and 3D cultures. In our past and current studies, we compared the ACTB expression levels between them using the cycle threshold (Ct) values; no significant differences were found. We briefly described it in Section 4.3 (Page 15, Lines 530-532 in the revised version).
- Line 821. The sentence needs to be reformulated. Cells (not PAX) were treated.
> We appreciate the reviewer’s careful reading of our manuscript. We amended the sentence (Page 7, Lines 241-242 in the revised version).
- Figure 8. Considering the regulatory role of Nrf2 activation on CYP3A4 expression, it could be expected that treatment with NK252 alone does increase CYP3A4 expression. However, this was not the case. Please, discuss.
>Similar to the reviewer’s comment, we predicted the increase in the CYP3A4 expression level by the treatment with Akt and Nrf2 activators (SC79 and NK252) in 3D spheroid models. However, as shown in Figures 6D-F, 8D-F, 10, and 11, the expression levels of CYP3A4 and CEBPB were not significantly changed by the single treatment with them. These results suggest that the Akt-Nrf2 signaling pathway may be almost sufficiently activated by the 3D spheroid formation. In accordance with the reviewer’s suggestion, we added this hypothesis in the ‘Discussion’ section (Page 13, Lines 426-428).

This manuscript is a resubmission of an earlier submission. The following is a list of the peer review reports and author responses from that submission.
Round 1
Reviewer 1 Report
Comments and Suggestions for Authors
The authors have presented their work on how down-regulation of CYP3A4 by the inhibition of KCa1.1 channel is responsible for overcoming the resistance to doxorubicin in cancer spheroids models. And by showing this they also show how cancer spheroids can be a relevant model for investigating doxorubicin resistance compared to 2D cancer models. Furthermore, they gave a mechanistic insight into how calcium-activated potassium channels KCa1.1 manipulate CYP3A4, essential in the metabolism of anti-cancer drugs. The study also offered insight into the Akt/Nrf/Nanog signaling pathway and its role in CYP3A4 regulation. The study is relatively well done and incorporating comments from the peer reviewers could enhance the quality of the study.
Comments for the authors -
1- could the authors please state what is the physiological relevance of the method of acquiring resistance to dox that happens in spheroids, could this be a phenomenon only observed in spheroids? and would this study mainly be called a 2D vs 3D cancer model study?
would there be any studies the authors can cite from the literature which show the relevance of using this model, like a similar increase in KCa1.1 expression or CYP3A4 expression level in patients, from previously published papers or RNA seq data? is the same cytochrome being increased in the patients that one sees in the spheroids?
Adding a few explanatory points on this subject could make the manuscript much more relevant to the field.
2- why did the authors choose the three cancer lines, could the authors estimate if a similar effect could be seen on other cancer lines also
3- Could the authors clarify if all the experiments that the authors have shown, is it 3 different cancer lines with each having 4 technical replicates?
if yes, were the experiments for three different cancer lines done at the same time or were the 4 or 5 replicates done at the same time
and if they saw a major difference between the 4 technical replicates, or do they always show the same trend,
4- Paxilline from the supplier was shown to be potent at 1.9 nM , (https://www.caymanchem.com/product/11345/paxilline) the authors have used 10uM, could they explain why a lower concentration was not preferred?
5- Could the Figure 11 A be explained why did LNCaP show an opposite trend compared to MG-63 or SW-1353
6- Figure 12 A and 12 B could be redone, and the figure legend could be improved and elaborated to comprise all the findings by the authors, if possible the schematic could include numbering for the steps taking place, and how the Dox is getting metabolized, and if possible how the CYP3A4 and MRP1/5 are interacting with dox
7- Could the authors kindly include a few points and estimate if this mechanism is only at a certain concentration of dox, or could there be certain patient subsets more susceptible to dox resistance, could there be certain SNPs involved in this? Could this be true for other types of anthracylines or other anti-cancer drugs where CYP3A4 is involved ? and explain dox metabolism in brief in patients
8- Line 331 & 342 , 343, , the block of K channels and change in the resting membrane potential decrese in Ca2+, these changes could have a serious side effect for electrcically active cells like cardiomyocytes, and could be arrhythmia inducing, have the authors considered this, can the find out and include the level of selectivity of the KCa1.1 block that it does not affect IKr and IKs. , if this is not relevant for human study, could they include an alternative?
9- could the authors explain if the siRNA knock-out done, how selective and successful was it and controls used in supplementary section.
10 - Could the authors explain what they included the taxanes, lines 391--395 and explain why they used ketoconazole vs other drugs,
Author Response
We would like to thank the reviewer for his/her valuable comments. We have attended to all the points raised by the reviewers. Each comment is highlighted below with our response underneath.
1-1. Could the authors please state what is the physiological relevance of the method of acquiring resistance to dox that happens in spheroids, could this be a phenomenon only observed in spheroids? and would this study mainly be called a 2D vs 3D cancer model study?
> As pointed out by the reviewer, it is important to indicate the physiological and clinical significance of 3D spheroid models examined in this study. This is a model study mimicking cancer stem-like cells. In the future study, we would like to clarify the clinical significance of this model study to collaborate with clinicians at our university.
1-2. Would there be any studies the authors can cite from the literature which show the relevance of using this model, like a similar increase in KCa1.1 expression or CYP3A4 expression level in patients, from previously published papers or RNA seq data? is the same cytochrome being increased in the patients that one sees in the spheroids? Adding a few explanatory points on this subject could make the manuscript much more relevant to the field.
> Both CYP3A4 and KCa1.1 are amplified in the patients and are biomarkers for predicting a poor prognosis and survival in cancer. We introduced this content in Refs. 2, 3, 8, and 9. However, it has not yet reported the relationship between cancer stemness and KCa1.1/CYP3A4 expression. In addition, we checked RNA seq data, which is available for free, and NCBI GEO Profiles; however, similar results were not found. In the 3D spheroid examined in this study, the KCa1.1 gene is not amplified and the KCa1.1 protein level is regulated by FBXW7-mediated protein degradation. The CYP3A4 gene is located on 7q22.1. Litviakov reported that stemness genes are located on 11.23, 21.13, 31.2, and 32.1 of 7q. We described that the chromosomal location is different between CYP3A4 (7q22.1) and stemness markers (11.23, 21.13, 31.2, and 32.1 of 7q) in the ‘Discussion’ section (Page 12, Line 391 - 393; Ref. 29). Thank you for your adequate indication.
Litviakov, N.; Ibragimova, M.; Tsyganov, M.; Kazantseva, P.; Deryusheva, I.; Pevzner, A.; Doroshenko, A.; Garbukov, E.; Tarabanovskaya, N.; Slonimskaya, E. Amplifications of stemness genes and the capacity of breast tumors for metastasis. Oncotarget. 2020, 11, 1988-2001. doi: 10.18632/oncotarget.27608.
2. Why did the authors choose the three cancer lines? Could the authors estimate if a similar effect could be seen on other cancer lines also?
> It has been reported that KCa1.1 is expressed in many types of cancer (colon, prostate, breast, renal, cervical, ovarian, pancreas, lung, renal, and glioma). Of over 15 cancer cell lines we possess, LNCaP (prostate cancer), MG-63 (osteosarcoma), SW-1353 (chondrosarcoma), and MDA-MB-453 (breast cancer) functionally expressed KCa1.1 K+ channel. In our previous study, MDA-MB-453 cells highly expressed KCa1.1; however, the aggregated spheroids of MDA-MB-453 were not formed by 3D culture. Therefore, we used three cell lines in the studies using 3D spheroids. Generally (also in ‘Cancers’ of MDPI), the usage of more than two cell lines is recommended in the study using cancer cell lines.
3. Could the authors clarify if all the experiments that the authors have shown, is it 3 different cancer lines with each having 4 technical replicates? if yes, were the experiments for three different cancer lines done at the same time or were the 4 or 5 replicates done at the same time and if they saw a major difference between the 4 technical replicates, or do they always show the same trend.
>As described by the reviewer, 4 replicate at the same time is NOT ‘n =4’. We prepared cDNAs and protein lysates from cells that were individually seeded on another day. We added the descriptions in Sections 4.3. and 4.4 (Page 14, Line 510 - 511; Page 14, Line 526 - 527).
4. Paxilline from the supplier was shown to be potent at 1.9 nM, (https://www.caymanchem.com/product/11345/paxilline) the authors have used 10 uM, could they explain why a lower concentration was not preferred?
> The indication by the reviewer is quite right. However, several reports have described that the IC50 for PAX is around 10 nM, but channels were largely closed to near 10 μM as maximal Po (open probability). Indeed, 1 μM PAX is usually used to inhibit KCa1.1 current in electrophysiological experiments. In the cell culture media, PAX can bind to supplemental serum. In our preliminary study, 1 μM PAX induced a small decrease in CYP3A4 transcription in the spheroid models examined.
5. Could the Figure 11 A be explained why did LNCaP show an opposite trend compared to MG-63 or SW-1353
> We are also interested in the results in Fig.11. However, we cannot explain why. We prepared the 3D spheroids from the other cancer cell lines (prostate cancer cell, PC3, colonic cancer cell, HCT-116, and breast cancer cell, YMB-1) that acquire DOX resistance by spheroid formation but do not express KCa1.1. YMB-1 showed a marked decrease in SLC22A16 level, similar to MG-63 and SW-1353; however, in PC-3 and HCT-116 spheroids, expression levels of SLC22A16 were very low. Therefore, the decrease in SLC22A16 expression is not common in DOX resistance-acquired cancer spheroid models.
6. Figure 12 A and 12 B could be redone, and the figure legend could be improved and elaborated to comprise all the findings by the authors, if possible the schematic could include numbering for the steps taking place, and how the Dox is getting metabolized, and if possible how the CYP3A4 and MRP1/5 are interacting with dox
> We really thank the reviewer for adequate suggestions such as ‘including numbering for the steps’. In accordance with the reviewer’s comment, we improved the figure and its legend (Fig. 13 in the revised manuscript) (Page 11, Line 366 – 374).
7. Could the authors kindly include a few points and estimate if this mechanism is only at a certain concentration of dox, or could there be certain patient subsets more susceptible to dox resistance, could there be certain SNPs involved in this? Could this be true for other types of anthracyclines or other anti-cancer drugs where CYP3A4 is involved? and explain dox metabolism in brief in patients
> We agree with the reviewer’s important indication. We suppose that KCa1.1 inhibition overcomes the resistance to other types of anthracyclines and other cancer drugs involving CYP3A4 and/or MRP1/5 in KCa1.1-positive cancer. However, we have no data on them. In addition, we would like to confirm our hypothesis using clinical items. In accordance with the reviewer’s comment, we added a brief explanation of DOX metabolism in vivo (Page 2, Line 65 - 66). In addition, we amended Fig. 13A. We really appreciate the reviewer’s suggestion.
8. Line 331 & 342 , 343, , the block of K channels and change in the resting membrane potential decrease in Ca2+, these changes could have a serious side effect for electrically active cells like cardiomyocytes, and could be arrhythmia inducing, have the authors considered this, can find out and include the level of selectivity of the KCa1.1 block that it does not affect IKr and IKs. , if this is not relevant for human study, could they include an alternative?
> For clinical usage of KCa1.1 inhibitor, the reviewer’s indication is important. I was engaged in K+ channel research in cardiac cells. It is well-known that KCa1.1 is not functionally expressed in the plasma membrane of cardiac myocytes. KCa1.1 inhibitors, paxilline, penitrem A, and iberiotoxin did not affect the outward K+ currents including IKD, IKr, and IKs; however, I have no data at hand. When we get clinical evidence to some extent in the future, we will examine the effect of KCa1.1 inhibitor on cardiac IKr and IKs currents (also membrane potential) in cardiac myocytes.
9. Could the authors explain if the siRNA knock-out done, how selective and successful was it and controls used in supplementary section.
> In Figure S2 (Fig. S1 in the original version), CYP2B6 and CYP3A4 expression levels are monitored in siCYP3A4- and siCYP2B6-transfected cells, respectively. Therefore, we amended it. In the ‘2D’ cultured cells, siRNA efficacy 48 hr after transfection is over 70 % (70 to 85 %). However, we did not show siRNA efficacy in 2D culture conditions because expression levels of CYP3A4 and CEBP isoforms are quite different between 2D and 3D. We considered shRNA and CRISPR, which may be adequate for the long-term blockade of target genes; however, we did not examine the experiments using them. As shown in Figure 3, PAX treatment decreased the expression levels of CYP3A4 transcripts by approximately 50 %. Therefore, a 50% decrease in CYP3A4 level may reflect the effect of PAX.
10. Could the authors explain what they included in the taxanes, lines 391-395 and explain why they used ketoconazole vs other drugs,
> As pointed out by the reviewer, we first thought that the readers may confuse when the taxane data are included in this manuscript and that the taxane data are not involved in its title. However, it is most famous that taxanes are metabolized by CYP3A4. In addition, we reported that resistance to taxane is overcome by the treatment with KCa1.1 inhibitor in MG-63 and SW-1353 spheroid models [Ref. 5]. If the reviewer recommends deleting taxane section, we will remove them in the next revision. Furthermore, we removed ketoconazole results in Fig. S5.
The entire manuscript is carefully edited by all authors again. We found several errors and amended them.

Reviewer 2 Report
Comments and Suggestions for Authors
This study revealed that LNCaP, MG-63 and SW-1353 spheroid models have higher expression level of CYP2B6 and CYP3A4, either siRNA knockdown or pharmacological inhibition of CYP3A4 leads cancer spheroid models to be sensitive to doxorubicin treatment, and inhibition of the KCa1.1 K+ channel can down-regulate the CYP3A4. Overall, the finding presented here is interesting, but the technique was mainly limited within qPCR, and a big improvement of the manuscript is still required before its acceptance.
Below are some specific comments.
1. In abstract, line 18-20, “…..by the inhibition of KCa1.1 through the transcriptional repression pf CCAAT/enhancer binding proteins…..”, but the authors only showed that transcriptional repression of CCAAT/enhancer binding protein isoforms by the inhibition of KCa1.1. it is bit far to state “through”.
2. In abstract, line 24-25, “through the Akt-Nrf2-CEBP signaling pathway”, again, the paper lacks enough evidence to state “through”.
Author Response
We would like to thank the reviewer for his/her valuable comments. We have attended to all the points raised by the reviewers. Each comment is highlighted below with our response underneath.
1. In abstract, line 18-20, “…..by the inhibition of KCa1.1 through the transcriptional repression of CCAAT/enhancer binding proteins…..”, but the authors only showed that transcriptional repression of CCAAT/enhancer binding protein isoforms by the inhibition of KCa1.1. it is bit far to state “through”.
> In order to identify the signaling pathway involving KCa1.1 inhibition-mediated CYP3A4 down-regulation, we first examined the effects of Akt inhibitor, AZD5363 (Fig. 5A-C), and Nrf2 inhibitor, ML385 (Fig. 7A-C). But such experimental strategies are not logical. Therefore, we performed the effect of activators of Akt and Nrf2 on PAX-induced down-regulation of CYP3A4 (Fig. 5D-F, Fig. 7D-F). Similar experiments were examined the PAX-induced down-regulation of CEBP isoforms (Fig. 9, 10). Generally, Nrf2 is downstream of Akt, so we do not show the results of the Akt-Nrf2 signaling pathway.
We here indicated the data for the reviewer only.
Figure. A-C: Real-time PCR examination of CYP3A4 in LNCaP (A), MG-63 (B), and SW-1353 (C) spheroid models treated (+) or untreated (−) with 2 μM AZD5363 (AZD) and 100 μM NK252 for 12 hr (n = 4 for each). D-F: Real-time PCR examination of CYP3A4 in LNCaP (D), MG-63 (E), and SW-1353 (F) spheroid models treated (+) or untreated (−) with 10 μM ML385 and 10 μM SC79 for 12 hr (n = 4 for each). After normalization to ACTB mRNA expression levels, CYP3A4 mRNA expression levels in −/− group were expressed as 1.0. **: P < 0.01 vs. −/−; ##: P < 0.01 vs. +/−.
Importantly, we recently identified that CEBPB but not CEBPA and CEBPD is a contributor of CYP3A4 transcription in LNCaP, MG-63, and SW-1353 spheroid models. Therefore, we added the results (Fig. 11 in the revised manuscript) and the descriptions in the ‘Abstract’, ‘Results’, ‘Figure legend’, ‘Discussion’, and ‘Conclusions’ sections (Page 1, Line 25; Page 5, Line 204 - 213; Page 10, Line 344 - 350; Page 13, Line 434 - 436; Page 15, Line 568 - 570; Supplementary Fig S4).
An Akt activator, SC79, and an Nrf2 activator, NK252, did not affect the down-regulation of CYP3A4 induced by the transfection of CEBPB siRNA in LNCaP spheroid model (data not shown).
2. In abstract, line 24-25, “through the Akt-Nrf2-CEBP signaling pathway”, again, the paper lacks enough evidence to state “through”.
>Please see the responses above.
>The entire manuscript is carefully edited by all authors again. We found several errors and amended them.

Reviewer 3 Report
Comments and Suggestions for Authors
Sasumu et al have examined the involvement of CYPs in the acquisition of DOX resistance. The authors should be commended for using 3 models of cancer for all experiments and for comparing 2D versus 3D growth.
Major comments
- siRNA efficiency of 50% is poor (Line 98). Have the authors considered shRNA or CRISPR of key targets?
- Ketoconazole (line 100) is not a specific CYP3A4 inhibitor. It's primary target is lanosterol 14-demethylase inhibitor (CYP51A1)
- I struggled with data presented in Figure 2 as important controls are missing such as no siRNA with doxorubicin, and siRNA alone with no treatment. Without these controls the data can no be interpretated.
- Figure 2 should also include dose response curves for KCZ and Doxorubicin treatment.
- Figure 5 D-F is missing SC79 only treatment
- Line 114 "Western blot analysis showed increased CYP3A4 expression" However Figure 3D-F clearly shows decreased CYP3A4 expression.
- For all figures there are too many parts ie A-J. Please add cell line name ontop of each graph for clarity instead of individuals letters and group cell lines according to treatment.
- For all experiments no reasoning as to timing of drug used or concentration provided. How do you know the concentrations used are on or off target? 10uM of PAX is extremely high, is this physiologically achievable? Timing of drug treatment also changes with no reasoning ie Figure 5 uses 10uM pax for 12hrs then figure 6 uses 2 hours of treatment.
- No mention of antibody dilution or incubation timing/temperature in the western blot methods section.
- No logical reason for conducting experiments provided other than another paper published a finding so we wanted to validate it.
- Figures are extremely repetitive, majority are bar graphs showing mRNA expression. Consider using other visual methods to display results ie colony formation assays for cellular growth, capas3 3/7 staining as a measure of apoptosis etc.
- Unclear what the final message of the paper is
Comments on the Quality of English LanguageFine, minor edits required
Author Response
We would like to thank the reviewer for his/her valuable comments. We have attended to all the points raised by the reviewers. Each comment is highlighted below with our response underneath.
1. siRNA efficiency of 50% is poor (Line 98). Have the authors considered shRNA or CRISPR of key targets?
> We can agree with the reviewer’s opinion. In the ‘2D’ cultured cells, siRNA efficacy 48 hr after transfection is over 75 % (75-85 %). However, we did not show siRNA efficacy in 2D culture conditions because expression levels of CYP3A4 and CEBP isoforms are quite different between 2D and 3D. We considered shRNA and CRISPR, which may be adequate for the long-term blockade of target genes; however, we did not examine the experiments using them. As shown in Figure 3, PAX treatment decreased the expression levels of CYP3A4 transcripts by approximately 50 %. Therefore, a 50% decrease in CYP3A4 level may reflect the effect of PAX.
2. Ketoconazole (line 100) is not a specific CYP3A4 inhibitor. It's primary target is lanosterol 14-demethylase inhibitor (CYP51A1)
> As pointed out by the reviewer, ketoconazole is not a ‘selective’ CYP3A4 inhibitor. We amended it to a ‘potent’ CYP3A4 inhibitor (Page 1, Line 22; Page 3, Line 106; Page 6, Line 239). Figure 2D-F are supporting experiments to Figure 2A-C (inhibition of CYP3A4 and CYP2B6 by specific siRNAs). The quantitative analysis of CYP51A1 was not examined because CYP51A1 is not a DOX-target enzyme.
3. I struggled with data presented in Figure 2 as important controls are missing such as no siRNA with doxorubicin, and siRNA alone with no treatment. Without these controls, the data cannot be interpreted.
> Of course, we have the data. In accordance with the reviewer’s suggestion, we showed the data in controls (Supplementary Fig. S1) (Page 3, Line 100 - 103).
4. Figure 2 should also include dose response curves for KCZ and Doxorubicin treatment.
> In our previous reports, we examined the effects of 0.1, 1, and 10 μM DOX on the viability of 2D monolayers and 3D spheroid models of LNCaP, MG-63, and SW-1353. 0.1 μM DOX was sensitive to 2D monolayers of MG-63 and SW-1353 but less sensitive to 2D monolayer of LNCaP. On the other hand, 10 μM DOX was sensitive to all 3D spheroid models. 1 μM DOX was sensitive to all 2D monolayers and insensitive to all 3D spheroid models. We are understanding the dose-response curve is essential for pharmacological analyses; however, we used a single dose (1 μM) DOX due to such reasons. We also examined the effects of 0.1-10 μM KCZ on the viability of 2D monolayers and 3D spheroid models. 10 μM KCZ was sensitive to all 3D spheroid models, and 0.1 μM KCZ showed different sensitivities to 2D monolayers. 2 μM KCZ was insensitive to all 3D spheroid models and showed similar sensitivities to all 2D monolayers. For this reason, we used a single dose (2 μM) KCZ in this study.
5. Figure 5 D-F is missing SC79 only treatment
> In accordance with the reviewer’s comment, we add the results of the effects of a single treatment with SC79 and NK252 on the CYP3A4 expression level (Page 4, Line 156 - 158; Page 4, Line 180 - 182; Page 4, Line 197 - Page 5, Line 199). Also, we briefly described the effects of a single treatment with SC79 and NK252 on the CEBP isoform expression level.
6. Line 114 "Western blot analysis showed increased CYP3A4 expression" However Figure 3D-F clearly shows decreased CYP3A4 expression.
> As pointed out by the reviewer, we amended it (Page 3, Line 119 - 121). We really appreciate the careful reading by the reviewer.
7. For all figures there are too many parts ie A-J. Please add cell line name ontop of each graph for clarity instead of individuals letters and group cell lines according to treatment.
> As pointed out by the reviewer, we added the individual cell line names in Figures. Thank you for your adequate suggestion.
8. For all experiments no reasoning as to timing of drug used or concentration provided. How do you know the concentrations used are on or off target? 10uM of PAX is extremely high, is this physiologically achievable? Timing of drug treatment also changes with no reasoning ie Figure 5 uses 10uM pax for 12hrs then figure 6 uses 2 hours of treatment.
> The indication by the reviewer about PAX concentration is right. However, several reports have described that the IC50 for PAX is around 10 nM, but channels were largely closed to near 10 μM as maximal Po (open probability). Indeed, 1 μM PAX is usually used to inhibit KCa1.1 current in electrophysiological experiments. In the cell culture media, PAX can bind to supplemental serum. In our preliminary study, 1 μM PAX induced a small decrease in CYP3A4 transcription in the spheroid models examined.
In addition, we measured the changes in the transcriptional levels 12 hr after the drug treatments in our study. On the other hand, we measured the changes in the phosphorylation levels of Akt 2 hr after the drug treatments in our study. Generally, in the peak levels of the changes in phosphorylation levels of target proteins including Akt were measured 1 to 2 hr after drug treatments. We briefly and clearly described it (Page 4, Line 160 - 162).
9. No mention of antibody dilution or incubation timing/temperature in the western blot methods section.
> In accordance with the reviewer’s comment, we add the antibody dilution information (Page 14, Line 528 - 531). We thank the reviewer for a careful reading.
10. No logical reason for conducting experiments provided other than another paper published a finding so we wanted to validate it.
> We have recently discovered that KCa1.1 is involved in the overcoming of chemoresistance using CSC-like spheroid models (Ref. 4, 5). In those reports, we determined that KCa1.1 inhibition overcomes DOX resistance by inhibiting the transcription of the drug-efflux transporters, MRPs. Then, we newly discovered the involvement of a drug-metabolizing enzyme, CYP3A4, in KCa1.1 inhibition-induced overcoming of DOX resistance in cancer spheroid models, and also identified the signaling pathway of it. These findings will provide valuable information to K+ channel researchers.
11. Figures are extremely repetitive, majority are bar graphs showing mRNA expression. Consider using other visual methods to display results ie colony formation assays for cellular growth, capas3 3/7 staining as a measure of apoptosis etc.
> We can understand the reviewer’s comment. However, in this study, our aim in the second half is to clarify the mechanism underlying KCa1.1 inhibition-induced decrease in CYP3A4 transcripts. We are planning the confocal imaging analysis of the KCa1.1 inhibition-induced decrease in the nuclear translocation of Nrf2. However, it will take a few months to select nice Nfr2 and P-Nrf2 antibodies and to obtain imaging results from three cancer spheroid models (vehicle-, PAX-, SC79-, NK252-, and combination-treated groups). In addition, we are examining the involvement of the Akt-Nrf2 signaling pathway in the FBXW7-mediated KCa1.1 protein degradation in LNCaP and MG-63 spheroid models. We would like to include the imaging data of the nuclear translocation of Nrf2 in a future paper.
12. Unclear what the final message of the paper is
> We agree with the reviewer’s comment that Section 5 is not focused on the main results of KCa1.1 inhibition-induced down-regulation of CYP3A4. In accordance with the reviewer’s comment, we amended the descriptions in Section 5 (see Section 5).
In addition, we recently identified that CEBPB but not CEBPA and CEBPD is a contributor of CYP3A4 transcription in LNCaP, MG-63, and SW-1353 spheroid models. Therefore, we added the results (Fig. 11 in the revised manuscript) and the descriptions in the ‘Abstract’, ‘Results’, ‘Figure legend’, ‘Discussion’, and ‘Conclusions’ sections (Page 1, Line 25; Page 5, Line 204 - 213; Page 10, Line 344 – 350; Page 13, Line 434 - 436; Page 15, Line 568 – 570; Supplementary Fig S4).
> The entire manuscript is carefully edited by all authors again. We found several errors and amended them.

Round 2
Reviewer 2 Report
Comments and Suggestions for Authors
the authors didn't address my comments well, the manuscript is still not organized in a logical way.
Reviewer 3 Report
Comments and Suggestions for Authors
Thank you to the authors for addressing some of the reviewers concerns. However, there are still major apparent issues with the revised version.
1. Figure 4: There are too many parts (A-R) to this figure. Authors should group either by cell line or target.
2. Figure 5: SC79 treatment alone still missing from figure
3. Figure 6: AKT densitometry required.
4. Figure 7/10: NK252 control alone missing from figure
5. Dose response curves for Dox and KCZ described in the authors reply should be included as supp data.